# Nephro- and Cardiotoxic Effects of Etoricoxib: Insights into Arachidonic Acid Metabolism and Beta-Adrenergic Receptor Expression in Experimental Mice

**DOI:** 10.3390/ph17111454

**Published:** 2024-10-30

**Authors:** Yahya F. Jamous, Badrah S. Alghamdi, Yazun Jarrar, Emad A. Hindi, Mohammad Z. Alam, Gamal S. Abd El-Aziz, Rabee F. Ibrahim, Refal Bakhlgi, Salha M. Algarni, Hanin A. AboTaleb

**Affiliations:** 1Vaccines and Bioprocessing National Center, King Abdulaziz City for Science and Technology (KACST), Riyadh 12354, Saudi Arabia; 2Department of Physiology, Faculty of Medicine, King Abdulaziz University, Jeddah 21589, Saudi Arabia; basalghamdi@kau.edu.sa (B.S.A.); htaleb0004@stu.kau.edu.sa (H.A.A.); 3Neuroscience and Geroscience Research Unit, King Fahd Medical Research Center, King Abdulaziz University, Jeddah 21589, Saudi Arabia; eahindi@kau.edu.sa (E.A.H.); mzalam@kau.edu.sa (M.Z.A.); rafalbakhlgi@hotmail.com (R.B.); salgarni0186@stu.kau.edu.sa (S.M.A.); 4Department of Basic Medical Sciences, Faculty of Medicine, Al-Balqa Applied University, Al-Salt 19117, Jordan; 5Department of Anatomy, Faculty of Medicine, King Abdulaziz University, Jeddah 22252, Saudi Arabia; dr_gamal_said@yahoo.com (G.S.A.E.-A.); aalabrahem2@kau.edu.sa (R.F.I.); 6Department of Medical Laboratory Technology, Faculty of Applied Medical Sciences, King Abdulaziz University, Jeddah 21589, Saudi Arabia; 7Department of Biochemistry, Faculty of Science, King Abdulaziz University, Jeddah 21589, Saudi Arabia

**Keywords:** arachidonic acid, cardiotoxicity, cytochrome P450s, etoricoxib, nephrotoxicity

## Abstract

**Background:** Etoricoxib is a widely used anti-inflammatory drug, but its safety profile concerning cardiovascular and renal health remains inadequately explored. This study aimed to assess the nephro- and cardiotoxic effects of etoricoxib in a murine model, with a focus on its impact on arachidonic acid-metabolizing enzymes and beta-adrenergic receptors associated with drug-induced toxicity. **Methods:** Thirty-five *BALB/C* mice were randomly assigned to five groups: control, low-dose etoricoxib, high-dose etoricoxib, low-dose celecoxib, and high-dose celecoxib (a well-known nephro- and cardiotoxic NSAID). The treatments were administered for 28 days, after which hearts and kidneys were excised for physical and histopathological analysis, and the expression of arachidonic acid-metabolizing enzymes (cytochrome P450s, lipoxygenases, cyclooxygenases) and beta-1 adrenergic receptor (*adrb1*) and angiotensin-converting enzyme (*ace2*) genes were quantified using quantitative reverse transcription PCR (qRT-PCR). **Results:** Etoricoxib administration resulted in dose-dependent nephro- and cardiotoxic effects. Renal histology revealed glomerular atrophy or hypertrophy and significant damage to the proximal and distal convoluted tubules, including epithelial flattening, cytoplasmic vacuolation, and luminal widening. Cardiac analysis showed disorganized muscle fibers and hyaline degeneration. These changes were associated with altered gene expression: the downregulation of *cox2*, *cyp1a1*, and *cyp2c29* in the kidneys and the upregulation of *cyp4a12*, *cox2*, and *adrb1*, along with the downregulation of *cyp2c29* and *ace2* in the heart. **Conclusions:** Etoricoxib induces nephro- and cardiotoxicity, marked by alterations in *arachidonic acid* metabolism and beta-adrenergic signaling pathways. The drug affects the expression of *arachidonic acid*-metabolizing enzymes and *adrb1* in the heart while downregulating *cox2* and other related enzymes in the kidneys. These findings underscore the need for caution when prescribing etoricoxib, particularly in patients with pre-existing renal or cardiac conditions.

## 1. Introduction

Non-steroidal anti-inflammatory drugs (NSAIDs) are extensively used for their analgesic, antipyretic, and anti-inflammatory properties. Their primary mechanism of action involves inhibiting cyclooxygenase (COX) enzymes—specifically *COX-1* and *COX-2*—that mediate the conversion of arachidonic acid into prostaglandins, which are involved in pain, inflammation, and fever responses [1]. NSAIDs are broadly categorized into non-selective inhibitors of both COX-1 and COX-2, selective COX-1 inhibitors, and selective COX-2 inhibitors [1]. While selective COX-1 inhibitors are associated with a higher risk of gastrointestinal toxicity [2], selective COX-2 inhibitors are linked to an increased risk of cardiovascular events compared to non-selective NSAIDs [3].

For instance, rofecoxib, a selective COX-2 inhibitor, was withdrawn from the market following reports of severe cardiovascular complications, including myocardial infarction and stroke [4]. Despite these concerns, other selective COX-2 inhibitors, such as celecoxib and etoricoxib, continue to be widely prescribed, particularly for chronic inflammatory diseases in countries like Saudi Arabia [5]. However, the cardiovascular safety profiles of celecoxib and etoricoxib have been less extensively studied. De Vecchis et al. conducted a meta-analysis on the cardiovascular risks associated with these drugs, concluding that while cardiovascular toxicity exists, the clinical benefits of celecoxib and etoricoxib often outweigh these risks [5].

Although the exact mechanisms by which NSAIDs exert their cardiovascular and renal effects are not fully understood, emerging evidence suggests a key role for arachidonic acid metabolism. Baracho et al. demonstrated that sub-chronic etoricoxib administration increased mean arterial pressure and hematocrit levels in rats, indicating its potential impact on cardiovascular function [6]. Similarly, Jarrar et al. found that various NSAIDs, including diclofenac, ibuprofen, mefenamic acid, and meloxicam, caused histopathological changes in the heart, kidneys, and liver, likely due to disruptions in the expression of arachidonic acid-metabolizing cytochrome P450 (CYP) enzymes [7]. Further research highlighted that high doses of rofecoxib induced ventricular wall thickening in the heart, associated with the upregulated expression of beta-1 adrenergic receptors (ADRB1) and arachidonic acid-metabolizing enzymes (COX, CYP450, and LOX) in the heart and kidneys [8].

Arachidonic acid-metabolizing CYP450 enzymes play a crucial role in cardiovascular regulation. Human CYP4A11 and CYP4F2 metabolize arachidonic acid into 20-hydroxyeicosatetraenoic acid (20-HETE), a vasoconstrictor that promotes platelet aggregation [9]. In contrast, CYP2C9 and CYP2J2 convert arachidonic acid into epoxyeicosatrienoic acids (EETs), which have vasodilatory and anti-inflammatory properties [9,10]. Notably, 20-HETE-producing CYP enzymes are upregulated following the administration of cardiotoxic drugs, such as doxorubicin [11], and in androgen-dependent hypertensive animal models, where 20-HETE synthesis increases while EET synthesis decreases [12].

In Saudi Arabia, etoricoxib is frequently prescribed for inflammatory diseases, yet its cardiotoxic potential has not been extensively evaluated [13]. Thus, this study aims to investigate the sub-chronic effects of etoricoxib on the expression of arachidonic acid-metabolizing enzymes (*CYP450*, *COX*, and *LOX*) and the *ADRB1* gene in the heart and kidneys of mice. Additionally, this study compares the cardiotoxic effects of etoricoxib to celecoxib, a well-known *COX-2* inhibitor associated with an altered expression of arachidonic acid-metabolizing enzymes and cardiovascular toxicity [7,8].

## 2. Results

### 2.1. The Effect of Etoricoxib and Celecoxib on Body Weight

The effects of different doses of etoricoxi*b* and celecoxib on body weight were calculated as the percentage of body weight gain. Two-way repeated measures ANOVA showed a significant effect of weeks (F (2.414, 68.21) = 11.04, *p* < 0.0001) but no significant effect of weeks × treatment (F (16, 113) = 0.6057, *p* = 0.8737) nor for treatment alone (F (4, 29) = 0.5766, *p* = 0.6818).

Figure 1—Further analysis using Tukey’s post hoc test revealed that the daily administration of different doses of celecoxib and etoricoxib did not affect the percentage of weight gain compared with the control group throughout the 4 weeks.

One-way ANOVA analysis showed no significant (*p* > 0.05) change in the heart and kidney relative weight in relation to the total body weight among all groups when compared with the control group (Table 1).

### 2.2. Histological Results of the Kidney

An examination of kidney sections of the control group (Figure 2A,B) showed the normal histological structure of the renal cortex that contained mostly the renal corpuscles (RCs) and proximal (PCTs) and distal convoluted (DCTs) tubules. Each RC was formed of a glomerulus and Bowman’s capsule with a capsular space in between. The Bowman’s capsule has an outer layer formed of flat epithelial cells. The glomerulus was formed of a tuft of anastomosing capillaries in between mesangial cells. PCTs were lined with tall cuboidal cells that showed acidophilic cytoplasm and possessed a brush border of tall microvilli that nearly fully filled the lumen. DCTs were less frequent than PCTs, being distinguished by their larger and clearly defined lumina and lack of the brush border.

In the low-dose celecoxib-treated group (Figure 2C,D), the examination showed that most of the renal corpuscles appeared regular and normal in structure. However, some glomeruli appeared shrunken, with dilated glomerular capillaries. Moreover, some of the proximal and distal convoluted tubules appeared more or less normal, while others showed some evidence of damage in the form of small cytoplasmic vacuolation, the loss of brush borders, and wider lumina. Furthermore, some wider interstitial spaces and interstitial hemorrhages were seen.

The examination of the high-dose celecoxib-treated group (Figure 2E,F) showed more changes in the renal cortex as compared to the low-dose treated group, where some renal corpuscles had atrophic glomeruli while in others the glomeruli appeared hypertrophied, with a reduction in capsular space. In addition, more proximal and distal convoluted tubules showed some evidence of damage in the form of a flattening of their epithelial lining, cytoplasmic vacuolation, a loss of brush borders, and wider lumina. Additionally, wider interstitial spaces and interstitial hemorrhages were detected.

The examination of the low-dose etoricoxib group (Figure 2G,H) revealed little changes in the renal cortex, where some renal corpuscles had a distorted appearance with dilated glomerular capillaries. Moreover, some of the proximal and distal convoluted tubules showed some evidence of damage in the form of cellular vacuolation, a loss of brush borders, and wider lumina. Additionally, some wide spaces and interstitial hemorrhages were detected.

The examination of the high-dose etoricoxib-treated group (Figure 2I,J) revealed a variable range of histopathology in the renal cortex, where some renal corpuscles had atrophic or hypertrophied glomeruli. In addition, more proximal and distal convoluted tubules showed damaged structure in the form of a flattening of their epithelial lining, cytoplasmic vacuolation, a loss of brush borders, and wider lumina. Furthermore, wider interstitial spaces, interstitial hemorrhages, and inflammatory cellular infiltration were detected.

### 2.3. Histopathological Scoring of the Kidney

The results of the histopathological scoring of the kidney in different groups are displayed in Table 2. As seen, mild scoring values were observed regarding the assessed renal parameters in low celecoxib and eterocoxib doses. Conversely, in high doses of these two drugs, moderate to severe scoring values were detected as compared to the control group, with higher scores in eterocoxib when compared to celecoxib.

### 2.4. Histological Results of the Heart

In the control group, the examination of the longitudinal and transverse sections of the ventricular wall (Figure 3A,B) showed normal architecture in the form of branching and anastomosing cardiac muscle fibers running in different directions with little interstitial spaces that contained some blood capillaries. These fibers showed poorly developed cross striation and acidophilic cytoplasm with elongated vesicular nuclei, usually centrally located.

In the low-dose celecoxib group (Figure 3C,D), the examination showed some changes in the form of waviness and a disarrangement of cardiac muscle fibers with wider spaces between them. Furthermore, some fibers showed hyaline degeneration.

In the high-dose celecoxib group (Figure 3E,F), there were more obvious alterations, where many cardiac muscle fibers demonstrated disorganization and fragmentation with wide interstitial spaces and congested blood vessels. Additionally, some cardiac muscle fibers showed pyknotic nuclei, while others showed hyaline degeneration.

In the low-dose etoricoxib group (Figure 3G,H), the examination showed some disorganization and wider spaces between cardiac muscle fibers, with some fibers showing hyaline degeneration. In the high-dose etoricoxib group (Figure 3I,J), the examination showed more disarrangement of cardiac muscle fibers with wide interstitial spaces. Some of the fibers had lost their nuclei, while others showed pale acidophilic cytoplasm with pyknotic nuclei. Moreover, some fibers showed more hyaline degeneration.

### 2.5. Histopathological Scoring of the Heart

The results of the histopathological scoring of the heart in different groups are displayed in Table 3. As shown, mild scoring values were observed regarding the assessed cardiac parameters in low celecoxib and eterocoxib doses. On the contrary, in high doses of these two drugs, moderate to severe scoring values were detected as compared to the control group, with higher scores in eterocoxib when compared to celecoxib.

### 2.6. Analysis of Gene Expression

#### 2.6.1. Gene Expression of Arachidonic-Metabolizing cyp450s in the Kidney

It is found that the low- and high-dose administration of celecoxib significantly decreased (*p* < 0.05) the mRNA expression of *cyp1a1* (Figure 4B) and *cyp2c29* (Figure 4C) in almost comparable levels in the mouse kidneys. In addition, low and high doses of etoricoxib significantly down-regulated (*p* < 0.05) the expression of *cyp1a1* in almost similar levels (Figure 4B), while only the high-dose administration of etoricoxib significantly down-regulated (*p* < 0.05) the expression of *cyp2c29* gene in the kidney of treated mice (Figure 4C).

#### 2.6.2. Gene Expression of Arachidonic Acid-Metabolizing cyp450*s* in the Heart

The expression of arachidonic acid-metabolizing cyp450 genes in the heart is illustrated in Figure 5A–D. We found that both celecoxib and etoricoxib upregulated the expression of the *cyp4a12* (Figure 5A) and *cyp2j5* (Figure 5D) genes in a dose-dependent manner. The expression of *cyp1a1* (Figure 5B) was upregulated significantly (*p* < 0.05) by more than three folds in the hearts after administration of low-dose celecoxib. Furthermore, high doses of celecoxib and low and high doses of etoricoxib significantly downregulated (*p* < 0.05) the expression of the *cyp2c29* gene in the mouse hearts (Figure 5C).

#### 2.6.3. Gene Expression of *ephx2* Gene in the Heart and Kidney

Figure 6A,B shows the expression of the mouse *ephx2* gene in the heart and kidney after the administration of celecoxib and etoricoxib. In the kidney, only low-dose celecoxib downregulated significantly (*p* < 0.05) the expression of *ephx2* by more than three folds (Figure 6A). In the heart, only high-dose etoricoxib downregulated significantly (*p* < 0.05) the expression of *ephx2* by more than two folds.

#### 2.6.4. Expression of a*lox12* and *cox2* Genes in the Kidney

Neither of the tested NSAIDs in this study affected the expression of *alox2* in the kidney of the mice (Figure 7A).

It is found that etoricoxib administration to the mice downregulated significantly (*p* < 0.05) the expression of the *cox2* gene in a dose-dependent manner (Figure 7B), while only the high-dose celecoxib downregulated significantly (*p* < 0.05) the *cox2* expression in the kidney of experimental mice.

#### 2.6.5. Expression of *alox12* and *cox2* Genes in the Heart

We found that the administration of celecoxib downregulated significantly (*p* < 0.05) the expression of the *alox12* gene in the heart in a dose-dependent manner (Figure 8A). Furthermore, both low- and high-dose etoricoxib downregulated significantly (*p* < 0.05) *alox12* expression in the heart in comparison with the expression of *alox2* in the heart of control mice (Figure 8A).

Regarding *cox2* expression in the heart, low and high doses of celecoxib, but not etoricoxib, induced (*p* < 0.05) the expression of *cox2* (Figure 8B).

#### 2.6.6. Expression of *adrb1* and *ace2* Genes in the Kidney

Figure 9 shows the expression of *adrb1* and *ace2* after the administration of celecoxib and etoricoxib. It is found that neither of the tested NSAIDs affected significantly (*p* > 0.05) on the expression of mouse *adrb1* (Figure 9A) and *ace2* (Figure 9B) in the kidney.

#### 2.6.7. Expression of *adrb1* and *ace2* Genes in the Heart

As shown in Figure 10A, only low-dose celecoxib and high-dose etoricoxib upregulated significantly (*p* < 0.05) the expression of the *adrb1* gene in the heart of experimental animals by more than three folds. Regarding the expression of *ace2* in the heart, it is found that both celecoxib and etoricoxib downregulated (*p* < 0.05) the expression of *ace2* in a dose-dependent manner (Figure 10B).

## 3. Discussion

NSAIDs are well-documented to induce cardio- and nephrotoxicity [14,15]. While numerous studies have explored the toxicity of NSAIDs on the heart and kidneys, there is a notable paucity of research specifically addressing the cardio- and nephrotoxic effects of etoricoxib, a commonly used analgesic and anti-inflammatory drug. This study aimed to investigate the impact of etoricoxib on the heart and kidneys in experimental mice and compare its effects with celecoxib, another NSAID known for its pathohistological and molecular impacts on these organs [7,8]. Our findings indicate that etoricoxib exhibits significant cardio- and nephrotoxic effects, as demonstrated through histological analysis and associated molecular alterations in arachidonic acid-metabolizing pathways and *adrb1* overexpression in the heart. These results suggest that caution is warranted when administering etoricoxib to patients with a high risk of cardiac and renal conditions. However, further clinical studies are needed to confirm these findings.

Weight and relative organ weight are frequently used indicators of drug-induced toxicity [16,17]. In this study, etoricoxib did not alter the total body weight of the mice. However, the relative weight of the heart increased significantly (*p* < 0.05) in a dose-dependent manner following etoricoxib administration, similar to the effects observed with celecoxib. Pathohistological analysis revealed that both etoricoxib and celecoxib caused hyaline degeneration, disorganization, and the widening of spaces between cardiac muscle fibers. Askar et al. previously noted that the COX2-selective inhibitor rofecoxib increased ventricular wall thickness [8]. These pathological changes may explain the increased relative heart weight observed with etoricoxib administration.

Our findings on the toxicity mechanisms of etoricoxib, particularly regarding histopathological changes, align with existing research that suggests its potential to cause cardiovascular alterations. Hamza et al., found that etoricoxib caused hyaline degeneration associated with pyknotic nuclei and myolysis. Intramuscular edema, congested capillaries, and partial necrotic changes were evident in the myocardium in the hearts of experimental animals [18]. However, regarding the gene expression of arachidonic acid metabolizing enzymes and the *adrb1* gene, there are no studies investigating the effect of etoricoxib on the expression of these genes. Accordingly, some of the molecular and histological patterns we observed in this study may suggest novel insights into etoricoxib’s mechanism of toxicity, which have not been extensively explored in earlier studies.

Hyaline degeneration, a replacement of muscle tissue with fibrous connective tissue, is often caused by hypoxia [13]. NSAIDs are known to inhibit prostaglandin formation, leading to vasoconstriction and reduced blood supply [19]. This mechanism could partly explain the hyaline degeneration observed in the hearts of mice administered etoricoxib and celecoxib.

Molecular analysis revealed that etoricoxib induced the mRNA expression of the 20-HETE synthesizing enzyme *cyp4a12* in a dose-dependent manner while decreasing the expression of EETs-synthesizing *cyp2c29* and upregulating *cyp2j5* in a dose-dependent manner. Notably, only the high dose of etoricoxib significantly downregulated *ephx2* expression. These results are consistent with previous reports that NSAIDs upregulate *cyp4a12*, which is associated with cardiac failure due to increased levels of the proinflammatory compound 20-HETE [20]. The upregulation of *cyp2j5* and downregulation of *ephx2* may represent a compensatory mechanism to counteract the increased expression of *cyp4a12*. Further studies measuring 20-HETE and EET levels in the heart following etoricoxib treatment are recommended to evaluate the overall impact on these metabolites.

Our study also found that both celecoxib and etoricoxib upregulated the mRNA expression of *adrb1* and downregulated *ace2* expression in a dose-dependent manner. This finding aligns with Askar et al.’s observation that *adrb1* expression increases following rofecoxib administration [8]. Elevated *adrb1* expression is linked to cardiac diseases and drug-induced cardiotoxicity [21,22]. Additionally, reduced *ace2* expression leads to increased angiotensin II levels, a vasoconstrictor associated with cardiovascular diseases such as hypertension and heart failure [23].

Interestingly, only celecoxib, not etoricoxib, significantly increased *cox2* expression in the heart. This suggests that different NSAIDs may have distinct biological effects due to their chemical structures [24].

In terms of renal effects, our study found that low-dose etoricoxib decreased relative kidney weight, whereas high-dose etoricoxib had no significant impact. Histological analysis revealed that both etoricoxib and celecoxib caused dose-dependent kidney damage, characterized by hypertrophied glomeruli, cytoplasmic vacuolation in tubules, a loss of brush borders, and wider lumina, along with inflammatory cellular infiltration. Glomerular hypertrophy indicates high intraglomerular pressure [25], while cytoplasmic vacuolation may result from ischemia [26]. The observed decrease in EETs-synthesizing *cyp2c29* after etoricoxib administration supports previous findings that NSAIDs downregulate *cyp2c29* and are associated with nephrotoxicity [7,27].

Finally, our study showed that etoricoxib downregulated *cox2* expression in the kidney. Given that cox2 plays a role in renal hemostasis [28], its downregulation might contribute to the nephrotoxic effects observed with etoricoxib.

Overall, this study underscores the importance of understanding the specific effects of NSAIDs like etoricoxib on cardiac and renal health and highlights the need for careful monitoring in patients at risk for such adverse effects.

## 4. Materials and Methods

### 4.1. Chemicals

Celecoxib powder with a purity > 99% (C9180), etoricoxib powder with a purity > 98% (E2470), a total RNA extraction kit (R1200), a universal reverse transcriptase-polymerase chain reaction (RT-PCR) kit (M-MLV, free Taq polymerase) (RP1105), and an SYBR Green PCR Mastermix (SR1110) were purchased from Solarbio (Beijing, China). The PCR primers were purchased from Integrated DNA Technologies (Coralville, IA, USA).

### 4.2. Experimental Animals

A total of 35 *BALB/C* male mice (6–8 weeks old, 30 ± 2 g) were obtained from the King Fahd Medical Research Center’s (KFMRC) animal house. Temperature, humidity, a cycle of twelve hours of light and darkness, and unlimited access to water and regular feed were all ideal settings for the mice. To provide a 12 h light/dark cycle, we used an automated lighting system in the facility where the experiment was conducted. The system was programmed to alternate between 12 h of light and 12 h of darkness each day, ensuring consistency throughout the study period. Animal handling, care, and treatment were conducted in accordance with the guidelines and regulations of the animal care and use committee (ACUC 22-1-5) of KFMRC. The Biomedical Ethics Committee of King Abdulaziz University approved the study’s protocol (Approval No. 69-22).

### 4.3. Experimental Protocol

The mice were randomized to one of the following five groups (n = 7): control (0.2 mL of vehicle cellulose orally (PO) daily), low-dose celecoxib (35 mg/kg/day), high-dose celecoxib (70 mg/kg/day PO), low-dose etoricoxib (10.5 mg/kg/day PO), and high-dose etoricoxib (21 mg/kg/day PO). The vehicle cellulose is an inert carrier without a pharmacological effect. The dosage form of the drugs was suspension. The tested medications’ doses corresponded to the daily therapeutic doses for humans and are based on the body surface area of the mice, as described previously [29], as follows: the human therapeutic dose of celecoxib (200 mg/day) was translated to low-dose and high-dose equivalents for mice, corresponding to 35 mg/kg/day and 70 mg/kg/day, respectively. Similarly, the human dose of etoricoxib (60 mg/day) was converted to 10.5 mg/kg/day (low dose) and 21 mg/kg/day (high dose) for mice. All mice were subjected to 7 days of adaptation followed by 28 days of treatment with medications. The weight of the tested mice was recorded weekly, and the percentage of weight gain was calculated as weight gain (%) = (new weight − initial weight/initial weight) × 100 [30,31]. The mice were sacrificed at the end of the experiment, and their hearts and kidneys were isolated and weighed relative to their body weight on the day of scarification.

### 4.4. Histological Analysis

The histological examination was carried out as described previously [32]. After sacrificing the mice, heart and kidney samples were obtained and washed with 0.9% normal saline before being fixed in 10% formalin for more than 24 h. The samples were then dehydrated by passing them through a graded series of alcohol followed by xylene. The heart and kidney tissues were then embedded in pure paraffin wax. The heart and kidney sections were stained with hematoxylin and eosin. Finally, the prepared sections were photographed using a Leica^®^ microscope equipped with a digital camera.

#### Histopathological Scoring of the Kidney and Heart

In order to perform a quantitative evaluation of the histopathological changes of the kidney and heart, ten digital images (at a magnification of X200 and H & E staining) were randomly analyzed using image analysis software (ImageJ Version 1.47, National Institutes of Health, New York, NY, USA) from each group of both organs in a blind manner. In the kidney, scoring was conducted for the following criteria: glomerular degeneration, tubular degeneration, inflammatory cell infiltration, interstitial hemorrhage, and interstitial spaces [33]. Meanwhile, in the heart, scoring was conducted for the following criteria: loss of striation, myocardial vacuolation, myocardial necrosis, inflammatory cell infiltration, focal areas of hemorrhage, and interstitial spaces [34]. The lesions were scored as negative (−), mild (+), moderate (++), and severe (+++).

### 4.5. RNA Extraction and cDNA Synthesis

Approximately 100 mg of heart and kidney samples were isolated from each experimental mouse. A total RNA extraction kit (R1200) (Solarbio, China) was used to extract the total RNA from the isolated samples following the manufacturer’s instructions. The extracted mRNA was then reverse-transcribed into cDNA using the universal RT-PCR Kit (Solarbio, China) in a reaction mixture containing 100 units of Moloney Murine Leukemia Virus (M-MLV) reverse transcriptase and free Taq polymerase (RP1105). The reverse transcriptase reaction was incubated at 37 °C for 60 min before being deactivated at 65 °C for 5 min.

### 4.6. Gene Expression Analysis

The mRNA expression of mouse beta-adrenergic 1 receptor (*beta1*), angiotensin converting- enzyme 2 receptor (*ace2*), cyclo-oxygenase (*cox*)1, *cox2*, lipoxygenase (*lox*)12, *lox*15, cytochrome (*cyp*)4a12, *cyp1a1*, *cyp2c29*, *cyp2j5*, soluble epoxide hydrolase (*ephx2*), and *β-actin* genes were examined in this study. The mouse *cyp4a12*, *cyp1a1*, *cyp2c29*, and *cyp2j5* genes are equivalent to the human *CYP4A11*, *CYP1A1*, *CYP2C9*, and *CYP2J2* genes, respectively [35]. The primer sequence and amplicon size are as published previously by Khirfan et al., (2022). The expression of these targeted genes was analyzed using qPCR, as previously described [32]. Briefly, 100 ng of the synthesized cDNA was added to a reaction mixture containing SYBR Green PCR Mastermix (SR1110) (Solarbio, China) and 10 pmol of forward and reverse primers. The PCR conditions followed specific steps: (1) Denaturation at 95 °C for 3 min, (2) 40 cycles of denaturation at 95 °C for 10 s, and (3) annealing at 53 to 58 °C (depending on the primer sequence of each targeted gene) for 30 s. The *β-actin* gene was used as a housekeeping gene in this study. The delta-delta Ct method was used to calculate the relative fold gene expression of samples [36,37].

### 4.7. Statistical Analysis

The change in the mRNA expression of the examined genes was expressed as a fold change compared to the control group. One- or two-way analysis of variance (ANOVA) test followed by Tukey analysis was used as the statistical tool to compare the control and other groups. The alteration in the expression of the tested genes was considered significant when the *p* value was <0.05. The Statistical Package for Social Sciences (version 26) was used as a statistical software program to analyze the results of this study.

## 5. Conclusions

In conclusion, this study provides a comprehensive analysis of the pathophysiological alterations associated with etoricoxib administration, focusing on its impact on the mRNA expression of key arachidonic acid-metabolizing enzymes, as well as adrb1 and ace2 in the heart and kidneys of mice. Our findings reveal significant insights into the mechanisms underlying nephro- and cardiotoxicity induced by etoricoxib. Specifically, etoricoxib administration led to dose-dependent alterations in the expression of enzymes involved in arachidonic acid metabolism, such as *cyp4a12*, *cyp2c29*, *cyp2j5*, and *ephx2*, as well as changes in the expression of *adrb1* and *ace2*. These molecular changes correlate with the observed histopathological damage and alterations in organ weight, highlighting the potential risks associated with etoricoxib use.

Our results underscore the need for caution when administering etoricoxib, particularly in patients with pre-existing cardiac or renal conditions. The observed pathophysiological and molecular alterations contribute to a better understanding of the drug’s toxic effects and suggest that targeted monitoring may be necessary to mitigate potential adverse outcomes. However, to translate these findings into clinical practice, further research, including clinical trials, is essential to validate these results and fully elucidate the risks and mechanisms associated with etoricoxib-induced nephro- and cardiotoxicity.

## Figures and Tables

**Figure 1 pharmaceuticals-17-01454-f001:**
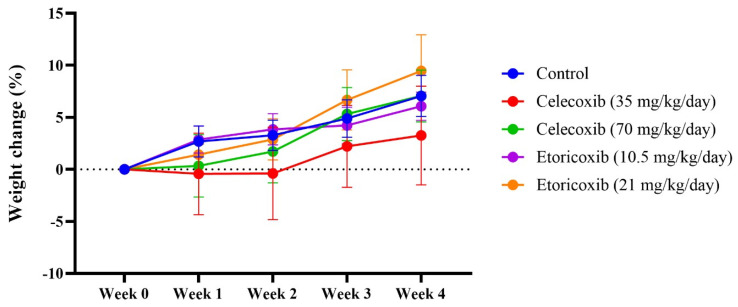
The effect of different doses of celecoxib and etoricoxib on the percentage change in body weight among groups. Data are presented as mean ± standard error of the mean (SEM). The effect of etoricoxib and celecoxib on the heart and kidney relative weight.

**Figure 2 pharmaceuticals-17-01454-f002:**
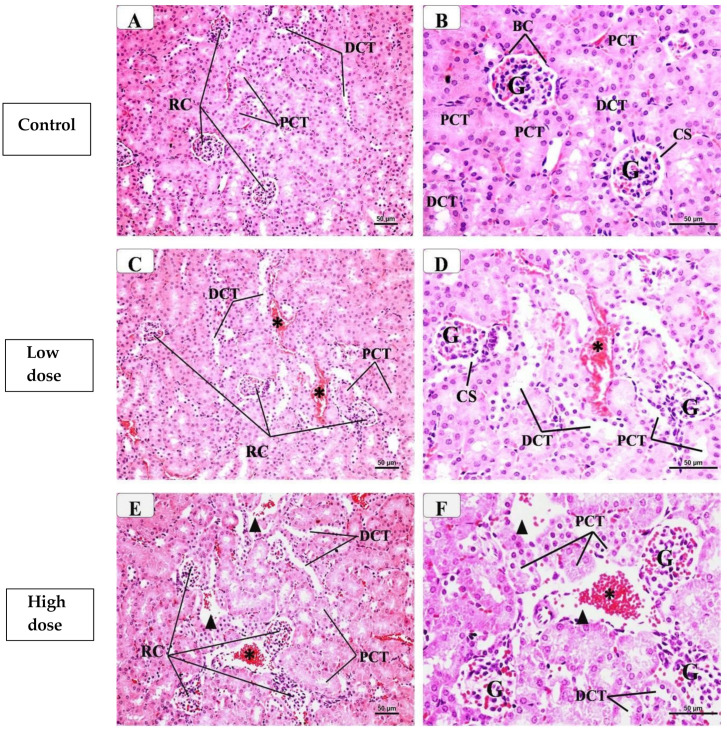
Representative photomicrographs of the H- & E-stained sections of the renal cortex from different groups after the administration of etoricoxib and celecoxib (H and E X 400). (**A**,**B**) represent the control group, with a normal structure containing renal corpuscles (RC) and proximal (PCT) and distal (DCT) convoluted tubules. The renal corpuscles consist of Bowman’s capsule (BC), which is lined by simple squamous and encloses the glomerulus (G) with normal capsular space (CS). The proximal convoluted tubules (PCT) are lined by tall cuboidal cells with acidophilic cytoplasm and a pale vesicular nucleus, while the distal convoluted tubules (DCT) are lined by short cuboidal cells with deeply stained nuclei. (**C**,**D**) represent the low-dose celecoxib group with some renal corpuscles (RC) showing atrophied glomeruli (G) and wide capsular space (CS). Some proximal (PCT) and distal convoluted (DCT) tubules have cytoplasmic vacuolation with dilated lumina. Notice the presence of some areas of interstitial hemorrhages (*). (**E**,**F**) represent the high-dose celecoxib group with many renal corpuscles (RC) having hypertrophied glomeruli (G) with decreased capsular space (CS). More proximal (PCT) and distal convoluted (DCT) tubules showed cytoplasmic vacuolation, a loss of brush borders, and wider lumina. Notice the presence of wide interstitial spaces (arrowhead) and areas of interstitial hemorrhages (*). (**G**,**H**) represent the low-dose etoricoxib group, showing some renal corpuscles (RC) having a distorted appearance with shrunken glomeruli (G) and wide capsular space (CS). In addition, some proximal (PCT) and distal convoluted (DCT) tubules have dilated lumina and a loss of brush borders. Notice the presence of some areas of wide spaces (arrowhead) and interstitial hemorrhages (*). (**I**,**J**) represent the high-dose etoricoxib group with some renal corpuscles (RC) having hypertrophied glomeruli (G) with decreased capsular space (CS). The proximal (PCT) and distal convoluted (DCT) tubules showed cytoplasmic vacuolation, a loss of brush borders, and wider lumina. Notice the presence of wide interstitial spaces (arrowhead), areas of interstitial hemorrhages (*), and inflammatory cellular infiltration (ICI).

**Figure 3 pharmaceuticals-17-01454-f003:**
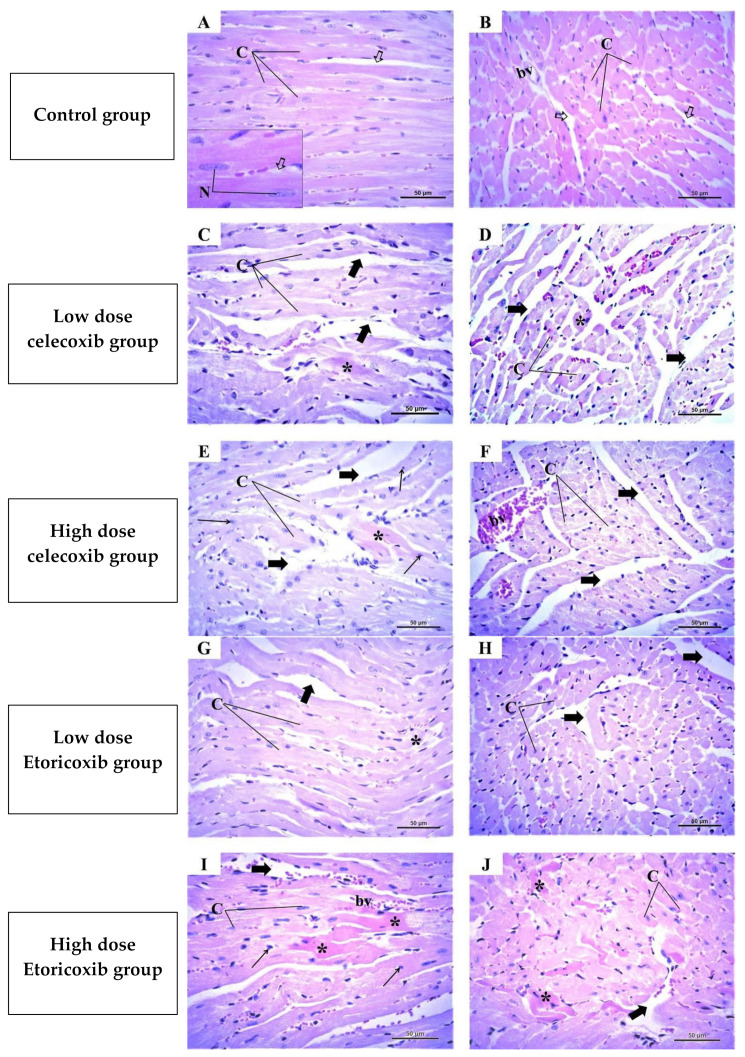
Representative photomicrographs of the ventricular wall of the hearts from different groups after the administration of etoricoxib and celecoxib (H and E X 400). (**A**,**B**) represent the longitudinal and transverse sections from the control group, with normal structure in the form of branched and anastomosing cardiac muscle fibers (C) which run in different directions with acidophilic cytoplasm and a poorly developed cross striation. Notice the presence of narrow interstitial spaces between the fibers that contain some nuclei of fibroblasts (🢧) and blood vessels (bv). The inset of high magnification showed elongated vesicular single nuclei (N) in cardiac muscle fibers. (**C**,**D**) represent the longitudinal and transverse sections from the low-dose celecoxib group with wavy and disarranged cardiac muscle fibers (C). Some fibers showed hyaline degeneration (*). Notice the presence of congested vessels (bv) and wide interstitial spaces (thick arrow). (**E**,**F**) represent the longitudinal and transverse sections from the high-dose celecoxib group with many cardiac muscle fibers (C) demonstrating disorganization and fragmentation with wide interstitial spaces (thick arrow) and congested blood vessels (bv). Notice that some cardiac muscle fibers showed pyknotic nuclei (thin arrow), while others showed hyaline degeneration (*). (**G**,**H**) represent the longitudinal and transverse sections from low dose etoricoxib group with some disorganization and wider interstitial spaces (thick arrow) between cardiac muscle fibers (C), with some fibers showing hyaline degeneration (*). (**I**,**J**) represent the longitudinal and transverse sections from the high-dose etoricoxib group with more disarrangement of cardiac muscle fibers (C) and wide interstitial spaces (thick arrow). Some of the fibers had lost their nuclei, while others showed pale acidophilic cytoplasm with pyknotic nuclei (thin arrow). Furthermore, some fibers showed more hyaline degeneration (*).

**Figure 4 pharmaceuticals-17-01454-f004:**
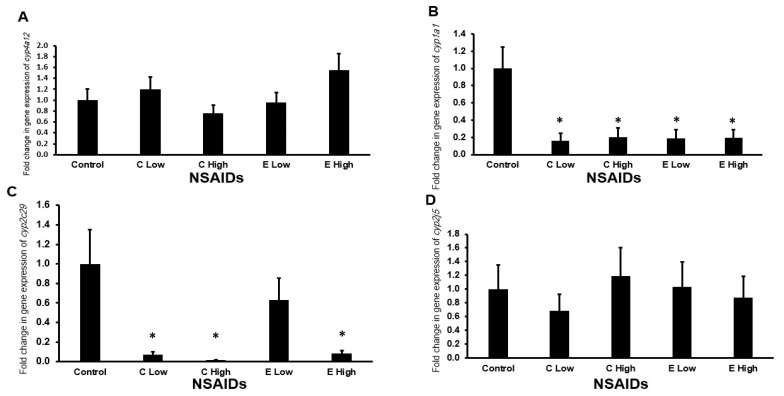
The mRNA expression of arachidonic acid-metabolizing cyp450 genes in the mouse kidneys after the administration of celecoxib and etoricoxib. The tested arachidonic acid-metabolizing *cyp450* genes are *cyp4a12* (**A**), *cyp1a1* (**B**), *cyp2c29* (**C**), and *cyp2j5* (**D**). “C” is the abbreviation of celecoxib, and “E” is the abbreviation of etoricoxib. “*” indicates statistical significance (*p* < 0.05, ANOVA) in comparison to the control group.

**Figure 5 pharmaceuticals-17-01454-f005:**
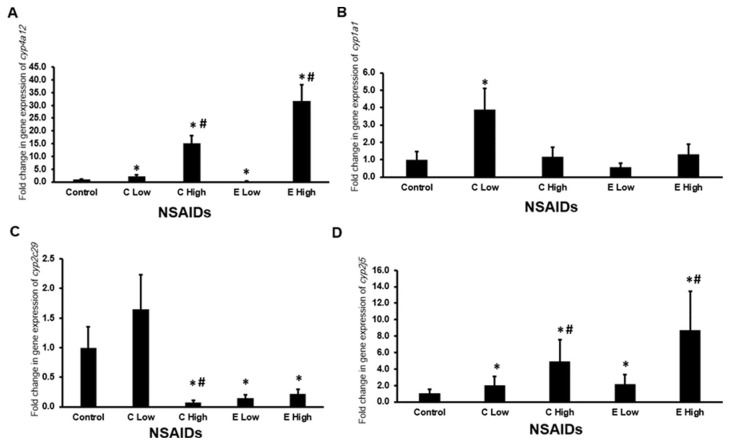
The mRNA expression of arachidonic acid-metabolizing cyp450 genes in the mouse hearts after the administration of celecoxib and etoricoxib. The tested arachidonic acid-metabolizing cyp450 genes are *cyp4a12* (**A**), *cyp1a1* (**B**), *cyp2c29* (**C**), and *cyp2j5* (**D**). “C” is the abbreviation of celecoxib, and “E” is the abbreviation of etoricoxib. “*” indicates statistical significance (*p* < 0.05, ANOVA) in comparison to the control group, while “#” indicates statistical significance (*p* < 0.05, ANOVA) in comparison with the low dose of the same tested drug.

**Figure 6 pharmaceuticals-17-01454-f006:**
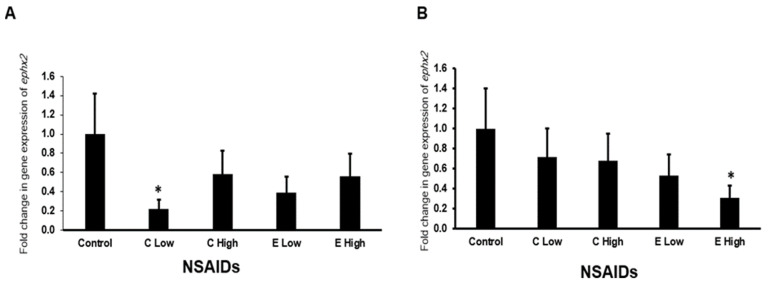
The mRNA expression of *ephx2* gene in the mouse kidneys (**A**) and hearts (**B**) after administration of celecoxib and Etoricoxib. “C” is the abbreviation of celecoxib and “E” is the abbreviation of Etoricoxib. “*” indicates the statistical significance (*p* < 0.05, ANOVA) in comparison to the control group.

**Figure 7 pharmaceuticals-17-01454-f007:**
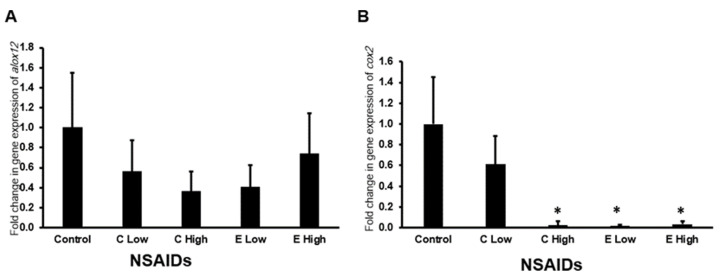
The mRNA expression of *alox12* (**A**) and *cox2* (**B**) genes in the mouse kidneys after administration of celecoxib and etoricoxib. “C” is the abbreviation of celecoxib, and “E” is the abbreviation of etoricoxib. “*” indicates statistical significance (*p* < 0.05, ANOVA) in comparison to the control group.

**Figure 8 pharmaceuticals-17-01454-f008:**
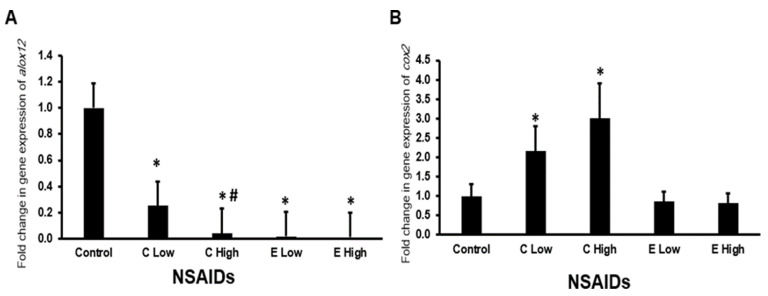
The mRNA expression of the *alox12* (**A**) and *cox2* (**B**) genes in the mouse hearts after the administration of celecoxib and etoricoxib. “C” is the abbreviation of celecoxib, and “E” is the abbreviation of etoricoxib. “*” indicates statistical significance (*p* < 0.05, ANOVA) in comparison to the control group, while “#” indicates statistical significance (*p* < 0.05, ANOVA) in comparison with the low dose of the same tested drug.

**Figure 9 pharmaceuticals-17-01454-f009:**
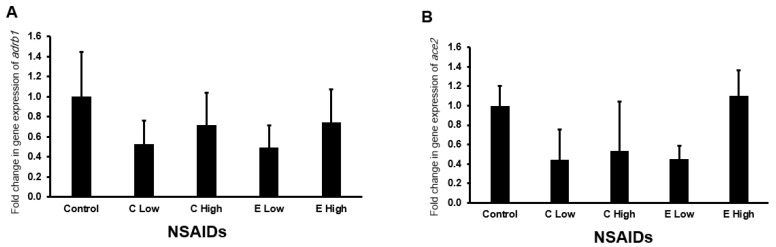
The mRNA expression of the *adrb1* (**A**) and *ace2* (**B**) genes in the mouse kidneys after the administration of celecoxib and etoricoxib. “C” is the abbreviation of celecoxib, and “E” is the abbreviation of etoricoxib.

**Figure 10 pharmaceuticals-17-01454-f010:**
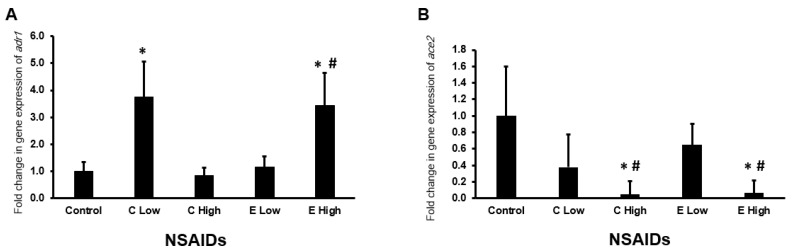
The mRNA expression of the *adrb1* (**A**) and *ace2* (**B**) genes in the mouse hearts after the administration of celecoxib and etoricoxib. “C” is the abbreviation of celecoxib, and “E” is the abbreviation of etoricoxib. “*” indicates statistical significance (*p* < 0.05, ANOVA) in comparison to the control group, while “#” indicates statistical significance (*p* < 0.05, ANOVA) in comparison with the low dose of the same tested drug.

**Table 1 pharmaceuticals-17-01454-t001:** Heart and kidney relative weight after administration of etoricoxib and celecoxib.

Groups	Heart (Weight %)	Kidneys (Weight %)
Control	0.533 ± 0.010	0.762 ± 0.021
celecoxib (35 mg/kg/day)	0.508 ± 0.022	0.788 ± 0.025 *
celecoxib (70 mg/kg/day)	0.605 ± 0.028 *	0.793 ± 0.025 *
etoricoxib (10.5 mg/kg/day)	0.563 ± 0.015 *	0.711 ± 0.022 *
etoricoxib (21 mg/kg/day)	0.553 ± 0.012 *	0.766 ± 0.031

Data are presented as mean ± standard error of the mean (SEM). “*” indicates statistical analysis (*p* < 0.05, ANOVA).

**Table 2 pharmaceuticals-17-01454-t002:** showing the histopathological scoring of the kidney in different groups.

Groups	Glomerular Degeneration	Tubular Degeneration	Inflammatory Cell Infiltration	Interstitial Hemorrhage	Interstitial Spaces
Control group	−	−	−	−	−
Low dose Celecoxib group	+	+	−	+	+
High dose Celecoxib group	++	++	++	++	++
Low dose eterocoxib group	+	+	+	+	+
High dose eterocoxib group	++	++	++	+++	+++

Lesions were generally scored as negative (−), mild (+), moderate (++), and severe (+++).

**Table 3 pharmaceuticals-17-01454-t003:** showing the histopathological scoring of the heart in different groups.

Groups	Loss of Striation	Myocardial Vacuolation	Myocardial Necrosis	Inflammatory Cell Infiltration	Focal Hemorrhages	Intramyocardial Spaces
Control group	−	−	−	−	−	−
Low dose Celecoxib group	+	+	−	−	−	+
High dose Celecoxib group	++	++	+	++	++	++
Low dose eterocoxib group	+	+	+	+	+	+
High dose eterocoxib group	++	++	++	++	+++	++

Lesions were generally scored as negative (−), mild (+), moderate (++), and severe (+++).

## Data Availability

The original contributions presented in the study are included in the article, further inquiries can be directed to the corresponding authors.

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
