# Peer review of "Nephro- and Cardiotoxic Effects of Etoricoxib: Insights into Arachidonic Acid Metabolism and Beta-Adrenergic Receptor Expression in Experimental Mice"

_pharmaceuticals, 2024, doi:10.3390/ph17111454_

Round 1

Reviewer 1 Report

Comments and Suggestions for Authors

Dear Authors,

Your manuscript is interesting and has the potential to be published, but I have certain suggestions and questions:

Please write unprotected drug names with a lowercase initial letter.

Are you familiar with the regulatory measures on the withdrawal of the drug from the market of other countries? Please give the consumption data of etoricoxib in Saudi Arabia (expressed as DDD/1000 population/day) for the last few years.

How did you provide "a cycle of twelve hours of light and darkness"? Please name the manufacturer of laboratory cages and other equipment.

Please clarify "vehicle cellulose".

Please provide more details about the dosage forms and the method of administration of the drugs used.

There is a problem with page numbering. At the bottom of the page, in the middle, below the text, there are numbers. Please correct it.

Good luck!

Author Response

Your manuscript is interesting and has the potential to be published, but I have certain suggestions and questions:

Please write unprotected drug names with a lowercase initial letter.

Reply: All unprotected drug names were written with a lowercase initial letter in the revised manuscript and highlighted in a yellow color.

Are you familiar with the regulatory measures on the withdrawal of the drug from the market of other countries? Please give the consumption data of etoricoxib in Saudi Arabia (expressed as DDD/1000 population/day) for the last few years.

Reply: Thank you for your valuable comment. We have carefully considered your suggestion to include the consumption data of etoricoxib in Saudi Arabia, expressed as DDD/1000 population/day. However, after conducting an extensive search through various databases and contacting relevant authorities (e.g., the Saudi Food and Drug Authority), we were unable to locate publicly available data on etoricoxib consumption specific to Saudi Arabia in this format.

Thank you for understanding.

How did you provide "a cycle of twelve hours of light and darkness"? Please name the manufacturer of laboratory cages and other equipment.

Reply: Thank you for your insightful comment. To provide a 12-hour light/dark cycle, we used an automated lighting system in the facility where the experiment was conducted. The system was programmed to alternate between 12 hours of light and 12 hours of darkness each day, ensuring consistency throughout the study period.

In response to the reviewer’s comment, we added these details in the revised manuscript, highlighted in yellow color.

Regarding the laboratory cages and other equipments, all are checked and the manufacturer names of the equipment’s were provided in the revised manuscript.

Please clarify "vehicle cellulose".

Reply: Thank you for your comment. In our study, "vehicle cellulose" refers to the cellulose substance used as a carrier for administering the drugs. It is an inert control substance without any pharmacological effects of its own.

In response to the reviewer’s comment, we clarified the vehicle-cellulose in the revised manuscript and highlighted in yellow color as followings:

The mice were randomized to one of the following five groups (n=7): control (0.2 ml of vehicle cellulose PO daily), low-dose celecoxib (35 mg/kg/day), high-dose celecoxib (70 mg/kg/day PO), low-dose etoricoxib (10.5 mg/kg/day PO), and high-dose etoricoxib (21 mg/kg/day PO). The vehicle cellulose is an inert solvent without a pharmacological effect. The tested medications' doses corresponded to the daily therapeutic doses for humans and are based on the body surface area of the mice, as described previously [14].

Please provide more details about the dosage forms and the method of administration of the drugs used.

Reply: the method of administration is orally (PO) and the dosage form is a suspension. These details are made clear in the revised manuscript.

There is a problem with page numbering. At the bottom of the page, in the middle, below the text, there are numbers. Please correct it.

Reply: This problem is corrected in the revised manuscript.

Reviewer 2 Report

Comments and Suggestions for Authors

The manuscript provides a detailed investigation into the nephro- and cardiotoxic effects of Etoricoxib, with a specific emphasis on arachidonic acid metabolism and beta-adrenergic receptor expression. The study offers valuable insights into the molecular and histopathological changes caused by Etoricoxib, and it draws comparisons with Celecoxib, a well-known NSAID with established toxicity. However, there are several minor revisions that should be made before the manuscript can be considered for acceptance in the journal Pharmaceuticals:

(1) While the manuscript references previous studies to support the chosen doses, a more explicit explanation of the rationale behind the dose selection, particularly in relation to human equivalent doses, would be beneficial. This would provide readers with a clearer context for understanding the study's findings.

(2) While the histopathological descriptions are detailed, incorporating quantitative measures, such as scoring systems for tissue damage, would offer a more objective and comparable assessment of the observed changes. This would strengthen the manuscript's conclusions and facilitate comparisons with other studies.

(3) The discussion section could be further enhanced by delving deeper into comparisons with existing literature, particularly regarding Etoricoxib's known effects. Are the molecular and histological findings presented in this study consistent with previous research, or do they reveal novel mechanisms of toxicity? A more in-depth analysis of the relationship between the study's findings and the broader literature would strengthen the manuscript's contribution to the field.

Comments on the Quality of English Language

The manuscript contains minor typographical and grammatical errors. A thorough proofreading is recommended to improve readability. 

Author Response

The manuscript provides a detailed investigation into the nephro- and cardiotoxic effects of Etoricoxib, with a specific emphasis on arachidonic acid metabolism and beta-adrenergic receptor expression. The study offers valuable insights into the molecular and histopathological changes caused by Etoricoxib, and it draws comparisons with Celecoxib, a well-known NSAID with established toxicity. However, there are several minor revisions that should be made before the manuscript can be considered for acceptance in the journal Pharmaceuticals:

(1) While the manuscript references previous studies to support the chosen doses, a more explicit explanation of the rationale behind the dose selection, particularly in relation to human equivalent doses, would be beneficial. This would provide readers with a clearer context for understanding the study's findings.

Reply:  Thank you for your valuable comment. We appreciate your suggestion to provide a more explicit explanation regarding the rationale for the chosen doses.

In our study, the doses of celecoxib and etoricoxib were carefully selected based on their human therapeutic equivalents. To ensure accurate dosing, we employed the method of allometric scaling, which converts human doses to mouse doses using the formula that accounts for differences in body surface area, as previously described in the referenced literature [14]. Specifically, the human therapeutic dose of celecoxib (200 mg/day) was translated to low-dose and high-dose equivalents for mice, corresponding to 35 mg/kg/day and 70 mg/kg/day, respectively. Similarly, the human dose of etoricoxib (60 mg/day) was converted to 10.5 mg/kg/day (low dose) and 21 mg/kg/day (high dose) for mice.

In response to the reviewer’s comment, the followings have been added in the revised manuscript in highlighted yellow color:

“The tested medications' doses corresponded to the daily therapeutic doses for humans and are based on the body surface area of the mice, as described previously [14], as followings: the human therapeutic dose of celecoxib (200 mg/day) was translated to low-dose and high-dose equivalents for mice, corresponding to 35 mg/kg/day and 70 mg/kg/day, respectively. Similarly, the human dose of etoricoxib (60 mg/day) was converted to 10.5 mg/kg/day (low dose) and 21 mg/kg/day (high dose) for mice.”

(2) While the histopathological descriptions are detailed, incorporating quantitative measures, such as scoring systems for tissue damage, would offer a more objective and comparable assessment of the observed changes. This would strengthen the manuscript's conclusions and facilitate comparisons with other studies.

Reply: Thanks to the reviewer for this important comment. In response to the reviewer, the scoring of the histological results were added to the revised manuscript as followings:

In Method part:

“Histopathological scoring of the kidney and heart

In order to perform a quantitative evaluation of the histopathological changes of the kidney and heart, ten digital images (at a magnification of X200 and H & E staining) were randomly analyzed using image analysis software (ImageJ Version 1.47, National Institutes of Health, USA). from each group of both organs in a blind manner. In the kidney, the scoring was done for the following criteria: glomerular degeneration, tubular degeneration, inflammatory cell infiltration, interstitial hemorrhage and interstitial spaces (Vardi et al., 2005; Toprak et al., 2008). While in the heart, the scoring was done for the following criteria: loss of striation, myocardial vacuolation, myocardial necrosis, inflammatory cell infiltration, focal areas of hemorrhage and interstitial spaces (Molh et al., 2008). The lesions were scored as negative (-), mild (+), moderate (++) and severe (+++).

Molh AK, Ting LC, Khan J, Al-Jashamy K, Jaafar H, Islam MN. Histopathological studies of cardiac lesions after an acute high dose administration of methamphetamine. Malays J Med Sci. 2008 Jan;15(1):23-30. 

Vardi N, Iraz M, Ozturk F, Ucar M, Gul M, Esrefoglu M, Otlu A (2005): Improving effects of melatonin on the histological alterations of rat kidneys induced by experimental diabetes. J. Inonu. Univ. Med. Fac. 12,145-152.

Toprak O, Cirit M, Tanrisev M, Yazici C, Canoz O, Sipahioglu M, et al. Preventive effect of nebivolol on contrast-induced nephropathy in rats. Nephrol Dial Transplant 2008;23(3):853-9.”

In Results:

 “Histopathological scoring of the kidney:

  • The results of histopathological scoring of the kidney in different groups were displayed in table X1. As seen, mild scoring values were observed regarding the assessed renal parameters in low celecoxib and eterocoxib Conversely in high doses of these two drugs, a moderated to severe scoring values were detected as compared the control group, with higher scores in eterocoxib when compared to celecoxib drug.
  •  

Table X1: showing the histopathological scoring of the kidney in different groups

Groups

Glomerular degeneration

Tubular degeneration

Inflammatory cell infiltration

Interstitial hemorrhage

Interstitial spaces

Control group

_

_

_

_

_

Low dose Celecoxib group

+

+

_

+

+

High dose Celecoxib group

++

++

++

++

++

Low dose eterocoxib group

+

+

+

+

+

High dose eterocoxib group

++

++

++

+++

+++

Lesions were generally scored as negative (-), mild (+), moderate (++) and severe (+++).

Histopathological scoring of the heart:

  • The results of histopathological scoring of the heart in different groups were displayed in table X2. As shown, mild scoring values were observed regarding the assessed cardiac parameters in low celecoxib and eterocoxib On the contrary in high doses of these two drugs, a moderated to severe scoring values were detected as compared the control group, with higher scores in eterocoxib when compared to celecoxib drug.

Table X2: showing the histopathological scoring of the heart in different groups

Groups

Loss of striation

Myocardial vacuolation

Myocardial necrosis

Inflammatory cell infiltration

Focal hemorrhages

Intramyocardial spaces

Control group

_

_

_

_

_

_

Low dose Celecoxib group

+

+

_

_

_

+

High dose Celecoxib group

++

++

+

++

++

++

Low dose eterocoxib group

+

+

+

+

+

+

High dose eterocoxib group

++

++

++

++

+++

++

Lesions were generally scored as negative (-), mild (+), moderate (++) and severe (+++). “

(3) The discussion section could be further enhanced by delving deeper into comparisons with existing literature, particularly regarding Etoricoxib's known effects. Are the molecular and histological findings presented in this study consistent with previous research, or do they reveal novel mechanisms of toxicity? A more in-depth analysis of the relationship between the study's findings and the broader literature would strengthen the manuscript's contribution to the field.

Reply: In response to the reviewer’s comment, we added the followings in Discussion, in a yellow highlighted color as followings:

“Our findings on the toxicity mechanisms of etoricoxib, particularly regarding histopathological changes, align with existing research that suggests its potential to cause cardiovascular alterations. Hamza et al., found that etoricoxib caused hyaline degeneration associated with pyknotic nuclei and myolysis. Intramuscular edema, congested capillaries, and partial necrotic changes were evident in the myocardium in the hearts of experimental animals. However, Regarding the gene expression of arachidnonic acid metabolizing enzymes and beta 1 gene, there are no study investigated the effect of etoricoxib on the expression of these gene. Accordingly, some of the molecular and histological patterns we observed in this study may suggest novel insights into etoricoxib’s mechanism of toxicity, which have not been extensively explored in earlier studies.”

Hamza RZ, Alaryani FS, Omara F, Said MAA, El-Aziz SAA, El-Sheikh SM. Ascorbic Acid Ameliorates Cardiac and Hepatic Toxicity Induced by Azithromycin-Etoricoxib Drug Interaction. Curr Issues Mol Biol. 2022 May 31;44(6):2529-2541. doi: 10.3390/cimb44060172. PMID: 35735613; PMCID: PMC9222074.